# Effect of Hybridization and Ply Waviness on the Flexural Strength of Polymer Composites: An Experimental and Numerical Study

**DOI:** 10.3390/polym14071360

**Published:** 2022-03-27

**Authors:** Sharath P. Subadra, Paulius Griskevicius

**Affiliations:** 1Department of Mechanical Engineering, Faculty of Mechanical Engineering and Design, Kaunas University of Technology, LT-51424 Kaunas, Lithuania; paulius.griskevicius@ktu.lt; 2Viezo, Kirtimu Str. 61b, LT-02244 Vilnius, Lithuania

**Keywords:** fibre-reinforced polymer composites, wind energy, composite stiffnesses degradation, numerical modelling

## Abstract

The study aims to ascertain the influence of hybridisation and ply waviness on the flexural behaviour of polymer composites. Two different resin systems, namely epoxy and Poly(methyl methacrylate)-PMMA, were chosen for the study, wherein two batches of carbon/glass hybrid composites (CGHC) were fabricated with the two resin systems. In addition to CGHC samples, four other neat batches with waviness (glass/epoxy and glass/PMMA) were prepared to study the effect of out-of-plane ply waviness. Two sets were additionally made with in-plane waviness (angles ranging from 15–35°) with epoxy to further understand the effect of waviness on flexural behaviour. Thereafter, two more batches of samples with neither waviness nor hybrid architectures were tested to achieve a better understanding of hybridization and the presence of waviness. It was seen that the hybridization of polymer composites introduces a pseudo-ductile behaviour in brittle composites, which makes the failure more predictable. An energy-based model was implemented to quantify the ductility introduced by hybridization. The presence of in-plane waviness increased the flexural load but reduced the modulus considerably. The presence of out-of-plane waviness decreased the flexural properties of composites drastically, though the displacement rate was seen to increase considerably. From the comparison between epoxy and PMMA, it was seen that PMMA exhibited similar flexural properties vis-à-vis epoxy. PMMA is easy to re-cycle and thus could serve as an ideal replacement for epoxy resin. Finally, a numerical model was built based on an LS-DYNA commercial solver; the model predicted the flexural behaviour close to what was seen in the experiments. The model could be calibrated correctly by ascertaining the influence of failure strain in the longitudinal direction, which is fibre dependent, and the failure strain in the transverse direction, which is matrix dependent.

## 1. Introduction

The demand for polymer composites has been on the upswing due to their light weight, damage tolerance, high specific strength, durability, maturity in processing, lower gas emissions, lower fuel consumption, etc., compared to metals [1,2,3]. The failure of polymer composites is sudden and catastrophic, owing to their brittle nature; thus, to ensure safe operations, higher safety factors are applied for components made from polymer composites. This could lead to over-designed components of composites, hence affecting their potential weight-saving benefits. Introducing ductility into a brittle material, i.e., achieving gradual failure [4], in composite structures could enhance their functionality, widening their application scope. 

The hybridisation of composite architecture has been accepted as an approach to introduce gradual failure in polymer composites [4,5,6,7,8,9,10,11,12]. This essentially includes combining low-strain materials (LSM) and high-strain materials (HSM) in an appropriate configuration. Though different possibilities of hybridising exists, the one most exploited is the inter-layer hybrid configuration, where mixing of different materials occurs on the ply level [8,9]. Extensive studies have been conducted on the hybrid effect exhibited by specimens under tension [13,14,15,16,17]. It has been found that thin carbon/epoxy pre-impregnated plies produced using tow spreading technology have suppressed damage mechanisms by obtaining lower energy release rates, delaying the propagation of intralaminar and interlaminar cracks [18,19,20,21,22]. This has introduced fragmentation of carbon plies as a new damage mechanism in composites, thus lending to gradual failure instead of catastrophic failure [4]. As the need arises to make polymer composites environmentally sustainable, alternative resin systems that are more recyclable need to be explored [23]. The hybrid effect in such resins needs to be confirmed and quantified. 

Fibre waviness is a common manufacturing-induced defect associated with thick composite structures. Two primary causes of waviness are the residual stress originating during curing and local buckling of fibres during filament winding [24,25,26,27,28,29]. Ideal properties associated with straight fibre materials are assumed when conducting structural analysis; this is a flawed practice, because these properties are either over-estimated or under-estimated in terms of the true experimental values (these take into account the presence of waviness) [30,31,32,33,34,35,36]. Loading of composite structures with ply waviness causes a three-dimensional stress state that can reduce their stiffness and strength, which is of particular concern in real-time applications such as wind turbines. Load-bearing parts with ply waviness in wind turbines, such as spar caps, lead to early global failure and kinking of the entire blade structure; thus ply waviness can cause structural problems, which must be considered in the analysis and design process of wind turbine blades [36]. An important parameter controlling the waviness-dependent properties is the wave amplitude and wavelength ratio (a/*λ*) [31,37,38,39]. Rai et al. [39] proved this theoretically, and the latter has been experimentally proven [24,25,26,27,28,29,30,31,32,33,34,35,36,37,38,39]. 

Making composite architectures to achieve gradual failure with flexural loading has not been an objective in many studies; maximising the flexural strength and modulus has been the concern [40,41,42,43]. Most flexural studies on hybrid composites [4] have identified the ideal combinations of different fabrics to achieve the hybrid effect, while quite few have quantified the hybrid effect achieved. Ductility Index is a mathematical term that can be employed to determine the energy expended during failure and hence better understand the damage propagation. While studying waviness in polymer composites, the current research has sought to understand the mechanical performance in tension, compression, and fatigue [44]. Several studies [24,25,26,27,28,29,30,31,32,33,34,35,36,37,38,39] have aimed to predict the strength and stiffness reduction in the presence of waviness. The effect of waviness on flexural strength has been less explored experimentally, Allison and Evans [45] studied the effect of waviness on flexural performance. The same study derived a failure criterion that could predict the load and location where failure will begin. Taking the lead from this research, the current study explores the effect of waviness on composites with two different resin systems. Matrix-dominated properties play an important role [46] in the presence of waviness; hence, this becomes the rationale to understand the role of different matrices while studying waviness. The study further explores a numerical model that predicts the flexural behaviour of hybrid composites and laminates with waviness. The numerical model is an attempt to explore the effect of carbon fabric on the hybrid effect and the matrix dominant properties in the presence of waviness. 

## 2. Experimental Methodology 

### 2.1. Materials and Design of Experiments

Uni-directional glass (areal density 220 g/m^2^) and carbon fabrics (areal density 120 g/m^2^) were supplied by R&G Faserverbundwerskstoffe GmbH (Waldenbuch, Germany). Epoxy resin based on Bisphenol A and its hardener (modified cycloaliphatic polyamine free of alkyl phenol and benzyl alcohol) was also sourced from the same firm. The methyl methacrylate (MMA: 617H119-Orthocryl Resin) resin and its polymeriser (Benzoyl Peroxide-BPO: Orthocryl resin 617P37, Otto Bock) were sourced from Otto Bock HealthCare Deutschland GmbH (Duderstadt, Germany). The composite architecture and the resin viscosity and density are elaborated in Table 1 and Table 2 respectively as per the data sheets from the manufacturer. The experimental work here was designed in four stages: (a) making of fibre/resin laminate composites, (b) investigation of flexural properties of the prepared panels and subsequent micro-structure, (c) estimation of the ductility of hybrid composites, and (d) validation of the numerical model. The experiments were carried out for 10 batches of specimens with different fibre architectures, resin, etc., as elaborated in Table 1, where T-3, T-4, T-7, T-8 were specimens with out-of-plane waviness, and T-9 and T-10 with in-plane waviness. T-3 and T-7 had waviness defined as concave up and T-4 and T-8 as concave down. 

### 2.2. Fabrication of Fibre/Resin Laminate Composites and the Flexural Test

The preparation of the panels was preceded by cutting the fabrics in accordance with the size of the panels; EN ISO 14125:1998/AC was followed for the preparation of specimens. Epoxy resin and its hardener solution with a ratio of 70:30 wt% of the total panel weight were mixed together using a mechanical mixer for 15 min, followed by keeping the solutions in a vacuum chamber at −100 bar for 10 min to eliminate air bubbles introduced during mixing. PMMA resin mix was prepared by mixing MMA monomer with BPO as an initiation system in the free-radical polymerization, with a weight ratio of 100:2 (MMA:BPO), using a mechanical mixer for 15 min; subsequently, the air bubbles were removed as before. The hand lay-up method was adopted to fabricate the panels, and in the case of introducing out-of-plane waviness into the laminates, a semi-circular die made of plastic was used. The plastic die had a length of 110 mm and diameter of 10 mm; the waviness angle obtained by the placement of this die was approximately 14 degrees. The fabric was placed over the die (wax was used a releasing agent on the die) one after the other, and the resin was spread over it by a roller; subsequently, the panels were vacuum bagged. This method gives repeatability in the results of post-flexural tests, as the method ensures a constant thickness of 1.9 ± 0.1 mm through the fabricating process. This can be achieved by thoroughly sealing the vacuum bag and maintaining the vacuum pressure constant by ensuring that there are no leakages. In-plane waviness was introduced by pushing the fabric upwards while immersed in resin. Thus, by doing so, T-9 was introduced with an in-plane wave angle of ~15°, and T-10 with an in-plane angle of ~35°. An infra-red lamp post-cure process was adopted (at 70 °C for 6 h), and the main curing process was carried out on an electronically controlled oven (temperature range 30–350 °C) (at 90 °C for 5 h). An automatic cutter was used to cut the specimens, and the cutting parameters were set as per the ISO standard adopted for this study. Five samples were assigned to each code mentioned in Table 1, and the accuracy of the cut specimens in terms of dimensions was satisfactory to obtain consistency in the experimental results. Later, 3-point bending tests were performed on a Tinius Olsen universal testing machine (UTM) having a maximum load capacity of 10 kN, within a span length of 60 mm and at a loading rate of 3 mm/min. 

### 2.3. Determination of Ductility of Specimens Subjected to Flexural Loading

Ductility of beams can be expressed in terms of a dimensionless ductility factor or DI based on the general curvatures, rotations, or reflections. However, this criteria is based on a yield and an ultimate strain found in ductile metals as opposed to brittle materials. Thus, an energy criterion was introduced to estimate the DI of brittle materials based on the consumed energy until failure [47,48,49,50]. Based on this framework, Naaman and Jeong (1995) [49] and Grace et al. (1998) [50] developed two different models to compute the DI, which is based on the total energy (E_total_), elastic energy (E_elastic_), and the failure energy (E_in-elastic_), as seen in Equations (1) and (2). E_total_ represents the area under the load-displacement curve up until the final failure, whereas E_elastic_ is defined as the area of the triangle formed at the failure load by the line having the weighted average slope of two initial straight lines of the load-displacement curve (Figure 1). Both methods gave accurate results, which served as a motivation for its use in calculation of ductility of brittle materials, such as concrete, [49,50]. Thus, both these methods were employed in this paper.
(1)$$DI\;\left(Naaman\right)=\frac{1}{2}\left(\frac{E_{total}}{E_{elastic}}+1\right)$$
(2)$$DI\;\left(Grace’s\right)=\frac{E_{in\text{-}elastic}}{E_{totol}}$$

### 2.4. Numerical Analysis of the Flexural Behaviour of All the Architectures Currently Studied

The finite element method (FEM) was adopted in the current work to understand the effect of hybridization of composite architecture and to ascertain the effect of waviness on composite flexural properties. The analysis was carried out using LS-DYNA software, which is classified as the most-used program for solving nonlinear problems using explicit time integration with precise results [51]. The use of LS-DYNA as a software was validated by its original developers (Livermore Software Technology Corporation, acquired by Ansys in 2019) for its generic applications. Thus, common applications of LS-DYNA include automotive, aerospace, metal forming and multi-physics problems [52]. Thus, in the context of the current research, a finite element model was to predict the flexural performance of composites with hybrid architecture and degradation of flexural properties in the presence of waviness. The modelling techniques require several parameters to be defined, and these include material properties, meshing size, loading conditions, and constraints. These parameters were defined and used as input for LS-DYNA modelling for each of the samples studied in this paper (Figure 2). 

In the material section, two composite material properties were defined for carbon and glass. The material properties were determined from static tests on the specimens and from the mathematical model developed based on the rule of mixtures, as elaborated in [53]. Thus, based on these methods, Table 3 gives an overview of the material properties used in this paper. The table has information on both glass and carbon composites with both resin types used. 

The composite plies were modelled using 4 Node Shell formulation, available in the LS DYNA Shape Mesher library. The mesh size (mesh type: square) was kept constant at 1 mm throughout the modelling. The size of the specimens was as per the ISO standard mentioned in Section 3.2. The number of layers in the model is as per Table 1. Regarding the boundary conditions, the specimens were constrained as a pin and roller [54] support. This implies a completely constrained motion in the z-direction and free in the y-direction (along the width), while in the x-direction (along the length), the specimens were fixed at one end and were allowed a translation motion at the other end.On defining the loading conditions, initially, a set of nodes on which the load would be applied were defined using the Boundary_SPC_SET option. Later, the loading curve was defined based on the actual experimental loading conditions, and the curve was assigned to the nodes through the option Boundary_Prescribed_Motion_Set.The composite failure was modelled using the material model MAT 54, which is a progressive failure model that uses the Chang–Chang failure criterion [55]. The model takes in 21 parameters that should be defined, 15 of which are physically based and 6 of which are numerical parameters. Among the 15 physical parameters, 10 are material constants; these are elaborated in Table 2. The remaining 5 parameters are tensile and compressive failure strain in fibre directions, the matrix and shear failure strains, and the effective failure strain. The 6numerical parameters were set at their default values. By conducting a parametric, study it was inferred that only DFAILT and DFAILM (DFAILT-Max strain for fibre tension, DFAILM-Max strain for matrix straining in tension and compression) needed to be adjusted. These terms and their explanations can be found in [51]. Adjusting the above two parameters helps simulate the tension/compression within the matrix between layers and the tension of fibres along the bottom of the specimen [51].

## 3. Results and Discussion

### 3.1. Flexural Characteristics of Composite Specimens: Pure, Hybrid, and with Waviness

Figure 3a,b shows the load displacement plots for the tested samples from both epoxy and PMMA. Since consistent plots were obtained among the samples studied, the presented plots are just for one sample set. It can be inferred that the PMMA and epoxy samples exhibited similar flexure response; though a significant increase in load was seen, it was generally comparable to epoxy specimens. A direct consequence of this outcome is the eventual replacement of epoxy with more recyclable PMMA. A closer observation shows the specimens with waviness tend to lose their flexural strength in comparison to those with no waviness in them. Though the specimens without any waviness/hybridisation tend to carry the highest flexural load, the failure post-maximum was abrupt. Hybrid specimens with both glass and carbon within the architecture failed gradually, with intermittent load drops, characterised as the hybrid effect. Though the maximum load is lesser when compared to pure glass specimens, the increasing complexity of composite damage mechanisms increased its ductility and hence reduced the abruptness of the damage, which is less desirable in real-time applications of these materials. The presence of waviness within the architecture is less desirable from a strength perspective; the reduction in flexural modulus was approximately 30% when compared to pure specimens in the case of epoxy and 36% in the case of PMMA. The presence of waviness hinders the normal damage mechanisms associated with composites when subjected to flexural loading. Most specimens with waviness failed abruptly, not because of any noticeable fibre failure, but due to delaminations in the waviness region. A noticeable increase in displacement was observed in specimens with waviness; this was noticed in the case of T-3 (epoxy) and T-8 (PMMA), which could be attributed to the geometry of the specimens. A microscopic inspection and observations are made in a later section in this research. 

In addition to out-of-plane waviness, the effect of in-plane waviness in the case of epoxy specimens was studied; PMMA was not considered in this case. As mentioned in the previous section, five sets of samples were subjected to flexure in each code, and Figure 3d is the mean of the results obtained. This was to avoid too much data and to keep the research more concise as to study the effect of waviness on the flexural behaviour rather than ascertaining material properties with different resin systems. As can be seen in the figure (Figure 3c–e), the introduction of in-plane waviness (angle ranging from 15° to 35°) had a profound influence on the flexural behaviour of composites. Even though a reduction in flexural modulus was observed as is the case with the introduction of waviness, an increase of 22% in load was observed between T1 and T9/T10. This increase was expected because, mathematically, when using classical laminate theory to ascertain the influence of in-plane and out-of-plane waviness on the various material properties (E_x_, E_y_, E_z_, G_xy_, G_xz_, G_yz_), an increase in G_xy_ is observed in the case of in-plane waviness; in the case of out-of-plane waviness, an increase in G_xz_ is observed [56]. In practice, the increase in the case of G_xz_ cannot be realised because of early interlaminar failure [56], as can be seen earlier in the case of the out-of-plane waviness specimens studied in this paper. As can be seen in Figure 3e, the ultimate failure in the case of T-1 and T-9 was due to fibre rupture, as is the case with most flexural tests. However, in the case of T-10, no fibre failure was seen, but the specimen lost its load-bearing capacity due to shear failure, which is the major reason behind de-lamination failure. This is in stark contrast to the failure seen in T-1 and T-9, where no shear failure was seen. A direct implication of the introduction of in-plane waviness is an increase in in-plane shear modulus. This could translate to a higher load- bearing capacity, where the load is equitably shared by both the reinforcing fibres and the resin used. To prove this hypothesis, further studies should be conducted in this regard. 

The results are indications that hybrid composites can be beneficial by altering damage mechanisms, though compromises on strength and stiffnesses are to be expected. Figure 4a,b below illustrates the maximum strength and flexural stiffness (EN ISO 14125:1998/AC), which can be obtained by the following equations:(3)$$\sigma_{F}=\frac{3PL}{2bd^{2}}$$
(4)$$E_{F}=\frac{L^{3}m}{4bd^{3}}$$
where *P, L, b, d,* and *m* are the maximum load, span, width, height, and initial slope from the load-displacement curve, respectively. The figure below is the average plot of the five samples studied in each code set, as there was consistency in the results obtained. As can be inferred from the figure below, a considerable drop in flexural strength was witnessed with the presence of waviness in the architecture. Though the drop in the case of the hybrid specimen (T-2) is negligible, waviness in T-3 and T-4 saw a considerable reduction in strength. Similar results are seen in PMMA samples, thus favouring them to replace epoxy for better recyclability.

While an alternative resin that is more environmentally friendly than epoxy is a better choice, the alternative should be as good as epoxy. Though PMMA was less stiff when compared to its epoxy counterparts, the strengths were comparable to epoxy in all cases studied. In addition to this, it was seen that the presence of waviness did reduce the strength of the specimens, but the percentage difference between the sample without waviness (T-5) and the one with waviness (T-7 and T-8) was approximately 20%. A considerable amount of reduction was seen in flexural properties while studying PMMA samples. Though T5 had lower modulus than T-1, the introduction of hybrid architecture increased the modulus significantly. Hybridising samples with PMMA considerably increased the ductility index (DI), which is indicative of an increment in the inelastic energy absorption capabilities of these samples; this will be detailed in the next section. When introducing PMMA as an alternative to epoxy, there is the drawback of making the structure more elastic in nature (more pliable). Thus, there exists a rationale to hybridise, resulting in a slight increase in stiffness, but an insignificant reduction in strength. Thus, to conclude, it can be ascertained that with some trade-offs, based on the requirements, an appropriate hybrid structure with correct distribution of carbon fabrics can be engineered with PMMA, as an alternative. The presence of manufacturing-induced damage like waviness and undulations seems to have lesser impact on strength loaded under flexure.

### 3.2. Energy Absorption and Estimation of Ductility Index of Samples under Flexure

The energy absorbed by specimens can be divided into two major components, the elastic and inelastic components, and these components quantify the ductility of composites. The inelastic energy is defined as the energy spent in damage initiation and propagation [49]. As mentioned in Section 2.3, the ductility index (DI) can be estimated from Equations (1) and (2) based on the calculated elastic and inelastic components. Figure 5A,B compares the ductility index estimated from both the equations for pure T1 and T5 samples against the hybrid T-2 and T-6. As ductility of any structure quantifies the ability to absorb inelastic energy without losing the loading capacity, a higher ductility would naturally signify a higher ability to absorb inelastic energy [47,48,49,50]. This is true mathematically, as ductility is directly proportional to the inelastic component [50]. Though the energy model proposed in [50] considers the total energy component, the inelastic part of this total energy cannot be discounted. Thus, based on these arguments, the T-2 and T-6, due to their hybrid architecture, fail in a controlled manner when loaded under flexure. The introduction of a single layer of carbon within multiple layers of glass introduced the hybrid effect considerably, as is evident from the figure below. The increase in ductility can be attributed to an increase in the inelastic component of the total energy, which was calculated to be 520 J in the case of T-2 and 861 J in the case of T-6. The hybrid effect can be attributed to the effect of interplay of low-strain carbon and high-strain glass fibres; the inter-laminar stresses also have an effect and cannot be neglected, as is evident from this study. Another important observation from the current study is that of the elastic component from the total energy. In the case of the non-hybrid specimens, the elastic component was the dominant energy absorption segment. This component does make the material brittle and less predictable, while the introduction of a hybrid architecture increased the inelastic component, reducing the elastic component to 40% of the total energy. Thus, in conclusion, hybridising of composites can reduce the unpredictability of composites considerably and make them an ideal choice for structural applications.

### 3.3. Analysis of Damage in the Composite Architecture Subjected to External Loading

As seen in the previous sections, T-2 and T-6 had comparable load-bearing capacity and higher ductility effect as opposed to T-1 and T-5; thus, it becomes imperative to check the damage at the microscale. An optical microscope was used to check the damage and infer the mechanisms leading to a higher ductility effect in T-2 and T-6. Therefore, T-1, T-5, T-2, and T-6 were examined to check the progression of damage.

#### 3.3.1. Epoxy Resin

The cross section of the specimens were analysed (in the case of epoxies: T-1 and T-2) using an optical microscope with 4 lens head with 4×, 10×, 40×, and 100× lenses, in addition to a 5MPx camera for image transfer to a PC. It must be noted that the waved architecture was omitted in the case of epoxy and PMMA from this study. This was because no noticeable ductility was observed in the case of specimens with waved architecture, apart from an increase in displacement to final load drop. This was true in the case of T-3 and T-8, where it can be seen (Figure 3) that the maximum displacement to final load was approximately 10 mm. It was observed that in the case of T-1 (Figure 6), when the load drops, the fabrics ruptured and there was considerable delamination on the other side of the loading. After this event, the specimens lost their load-bearing capacity and ultimately failed completely. However, in the case of hybrids (T-2 in Figure 6), the carbon fabric was still intact. Therefore, a combination of rupture of high-strain fabric such as glass, and no rupture in a high-strength fabric such as carbon, contributes to the hybrid effect, which introduces ductility into composites and makes their failure more predictable.

#### 3.3.2. PMMA Resin

The damage observed in the case of T-5 and T6 (Figure 7) was clearer, as there was less reflection, as was the case in T-1 and T-2. Nevertheless, the damage observed here is similar to that seen in the previous case. T-5 failed by complete rupture of the glass fabric on the opposite side of the loading, though no de-lamination was observed. T6 exhibited the hybrid effect due to the reasons mentioned in the previous paragraph, but no delamination was observed among the fabrics in this case. The carbon fabric remained intact, which could have contributed to a higher ductility index as compared to T-2; in addition, the effect of matrix cannot be neglected. It is in this context that the effect of “bending–stiffness mismatch” [57] plays an important role in contributing to the ductility effect.

Bending–stiffness mismatch could be attributed to several reasons, including differing material properties, stacking sequence, and ply thickness. It was observed in [58] that delamination along the thickness direction was caused by differing stiffness among the plies, rather than viewing them in the context of stress distribution. It has to be noted that while considering this hypothesis in the context of the hybrids studied here, no observable delamination was observed; it was the systematic rupture of low-strength glass fabric with an intact carbon in the middle that contributed to a differing bending–stiffness mismatch in the architecture. This mismatch in properties among the plies could be one of the major factors contributing to the ductility effect, though this has to be verified analytically using the classical laminate theory and additional experiments with differing thickness of glass and carbon fabrics.

### 3.4. Numerical Results

The numerical modelling approach elaborated in Section 2.4 was able to generate results that could capture the experimental behaviour with good compatibility. This compatibility is illustrated below in Figure 8 and Table 4, where the first three rows bearing columns marked *a, b, c,* and *d* and the last three rows bearing columns marked *e, f, g,* and *h* are dedicated for the epoxy (T-1, T-2, T-3, and T-4) and PMMA (T-5, T-6, T-7, and T-8), respectively. It can be noticed that the modelling approach was able to capture the load displacement in the elastic region with good precision. In the first row below, the stress field along the x-direction (along the length) is elucidated, and the second row elucidates the stress field in the xy-direction; similarly, the stress fields are elucidated for PMMA in fourth and fifth rows, respectively. All the figures below were captured at the same moment (steps) to adhere to a uniformity in results.

From the stress fields, it was observed that T-4 and T-8 did not have a stress concentration along the mid-span (where the load is applied), while the same could be seen for all other types. It was also seen that T-4 and T-8 had the maximum displacement before failure. An inference on these two observations could be that the actual span length of these specimens is large if the waviness region is imagined to be a straight line. That would make the specimens naturally more elastic in nature than those without any waviness. In addition to this, the same advantage can be realised in PMMA, while in the case of epoxy, a reduction in strength was observed with the introduction of waviness. In the case of PMMA, there was an observable reduction in strength when compared to straight counterparts, but this was not drastic. These could be simulated on LD DYNA using the MAT 54 model, as this keyword uses strain-based criteria to arrive at failure; especially in the case of flexure, DFAILT and DFAILM plays an important role.

Failure strains DFAILT and DFAILM are calculated by dividing the material modulus by their strengths, i.e., $\mathrm{DFAILT}=\frac{\mathrm{XT}}{E_{1}}$ and $\mathrm{DFAILM}=\frac{\mathrm{YT}}{E_{2}}$. In this study, a parametric study was carried out to determine the correct value, and a range between 0.01 to 0.05 was found to give good compatibility with experiments; higher values were found to over-estimate the failure load to a large extent. It was also found that DFAILC, which is the failure strain in the compressive direction, had some influence on the results, and its value was taken in the range of −0.01 to −0.03 ($DFAILC=\frac{XC}{E_{1}}$). Other parameters in MAT 54, such as FBRT, TFAIL, DFAILS, SOFT, and YCFAC, were given default values. With these parameters, MAT 54 could be an ideal material model to study the flexural behaviour of composites within any kind of architecture. The next section checks the sensitivity of the model to DFAILT and DFAILM, in the range as shown in Table 3. T1 was chosen for the study, as the parameters for glass composite needed to be adjusted; this avoided the complication involved when hybrids are considered, as the parameters for carbon must be changed as well.

#### Sensitivity of the Model to Different Modelling Parameters

Figure 9 elucidates the theoretical and numerical strain values that were subsequently used for the parametric study in this section. The theoretical values were obtained using the rule of mixture model available in the literature [58]. The theoretical strains seen in Figure 8 can be determined by knowing the material properties of the fabric and the resin used. Thus, the glass fabric stiffness was taken from literature [59] and is listed in Table 5 along with that of the resin (epoxy) used for the current study. The resin properties were ascertained through quasi-static tests, as per ISO 527(2). The aim here was to arrive at the correct calibration and hence to ascertain the influence of the respective parameters on the flexural response, which is otherwise difficult to obtain experimentally. With the theoretical values available, it gives an idea of how much the numerical model can be calibrated and its accuracy. From the figure below (Figure 9), a DFAILT of 0.048 and DFAILM of 0.1 was used in the parametric study (PS-2 and PS-6, respectively).

Figure 10 elucidates the influence of MAT 54 parameters DFAILT and DFAILM on the behaviour of T1 under flexure. Shear stress distribution was chosen in this study because it was found that DFAILM had an influence on the failure. A parametric study in this case is necessary to benchmark a numerical model to further validate its future use in similar applications. Thus, in this study, it was found that DFAILT and DFAILM influence the model outcome to a greater extent. The range of different parameters adopted in this study is shown in the Table 6. Six parametric studies (PS) were conducted on T1, since the rest should show similar results with the same set of parameters and to avoid the accumulation of extensive results, which would be harder to analyse. From this study, it can be inferred that material properties in the longitudinal and transverse directions to a greater extent influenced the flexural response of the composite beam. From the six parametric studies, PS-1 and PS-6 gave responses that closely resembled that of the experimental ones. PS-2 and PS-4 slightly overestimated the response, and as can be seen in Table 3, DFAILM was constant at 0.04 and DFAILT was in the range of 0.048–0.1. It should be noted that DFAILM at 0.1, obtained theoretically, calibrated the model similarly to that of the experimental response. PS-3 and PS-5 underestimated the response to a greater extent, wherein lower strain values were adopted (0.009). As was explained in the previous section, DFAILT and DFAILM are max strain for fibre tension and max strain for matrix straining in tension and compression, respectively; parameters related to fibre and matrix play an important role in the flexural response. In Section 3.1, the effect of in-plane waviness was found to increase the flexural strength and modulus. As seen in this section, an increase in the matrix strain had a profound influence in the flexural response. Relating this study to in-plane waviness could highlight the importance of flexural modulus and strength, as seen in Section 3.1.

Figure 10b shows the effect of an over-estimated DFAILT and DFAILM at 0.1. At 0.1, the failure strain is very high, and this makes the composites very stiff in the elastic region; as seen from Figure 10a,d, shear stress distribution, the element deletion that signifies failure, was not observed. It was observed that a lower strain than 0.015 in this regard was detrimental to the result, and hence it was not considered. Strains lower than 0.01 showed instabilities in the model, with non-uniform element failure. DFAILT at 0.009 saw the specimen fail (Figure 10f) pre-maturely (evident from the element deletion) and be less stiff than the experimental plots. The ideal strains that reflected the experimental results were 0.033 and 0.04 (Figure 10e), and hence these should be considered. These strains are the baseline values at which the model predicts the failure of the specimens correctly for the given material properties. It was noted that these strains are dependent on a variety of factors having direct co-relation to the material properties, such as specimen thickness, fibre volume fraction, etc.

## 4. Conclusions

In the current study, an attempt to ascertain the influence of hybridisation and ply waviness on the flexural behaviour of polymer composites was carried out. Epoxy and PMMA was chosen for the study, and hence 10 batches of specimens where cut and tested (each batch was named, from T1–T10, with specimens each). Based on the study, the following conclusions were drawn:-PMMA was found to have similar flexural strength to that of epoxy, though the flexural modulus was found to be lower. Hybridising the architecture did not alter the modulus, but a drop in strength was observed in the case of epoxy specimens. In the case of PMMA, hybridisation increased the modulus, but the increase was not significant, and the strength did not change significantly. Thus, from a strength perspective, PMMA could be a good alternative for epoxy, thus making composites more recyclable.-The presence of waviness was found to be detrimental in both epoxy and PMMA specimens; in the case of the former, there was severe reduction in strength and modulus. However, the presence of in-plane waviness was found to increase the load significantly; thus, waviness could have some positive effects on composites.-Hybridisation introduces ductility into composites, and this can be quantified using an energy-based model. Thus, it was observed that hybridised specimens (T2 and T6) exhibited higher ductility when compared to their purer counterparts. A level of 60% ductility was seen in T2 and T6, while in T1 and T5, it was abysmally low.-The hybrid effect was further studied using an optical microscope, and it was observed that the carbon fabric was still intact, without failure. The hybrid effect was introduced by a controlled failure of first the glass fabrics and subsequently the carbon. Bending–stiffness mismatch was another reason for this observation, though this must be studied further using the classical laminate theory.-Numerical models were built on LS DYNA using the material model MAT 54, available in the LS DYNA MAT library. The modelling approach selected was found to predict the flexural behaviour similar to experiments. Tensile strain-to-failure (DFAILT) and matrix strain-to-failure (DFAILM) was seen to influence the modelling outcome proportionately, and hence a parametric study was conducted to establish the correct values of DFAILT and DFAILM.

## Figures and Tables

**Figure 1 polymers-14-01360-f001:**
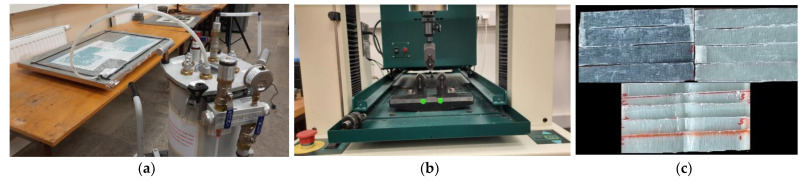
(**a**) Composite panel preparation using the vacuum bagging method. (**b**) Flexural test on a specimen with waviness. (**c**) Specimens for flexural test.

**Figure 2 polymers-14-01360-f002:**
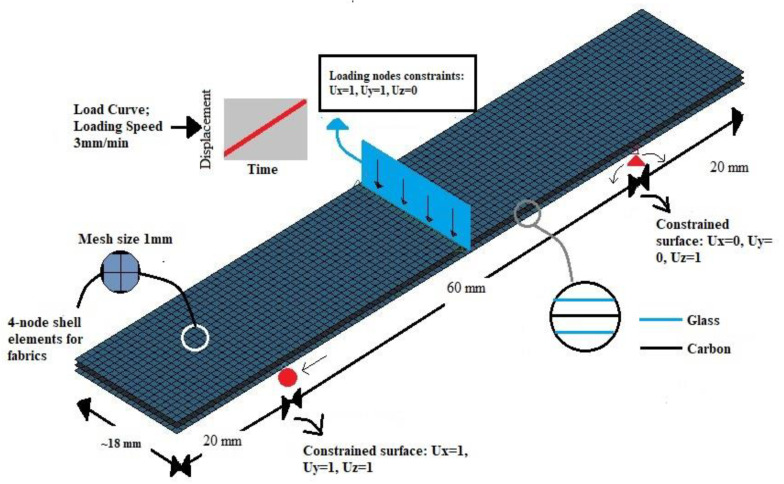
FEM schematic of the specimens, including modelling, materials, meshing, constraints, contact definition, and loading conditions.

**Figure 3 polymers-14-01360-f003:**
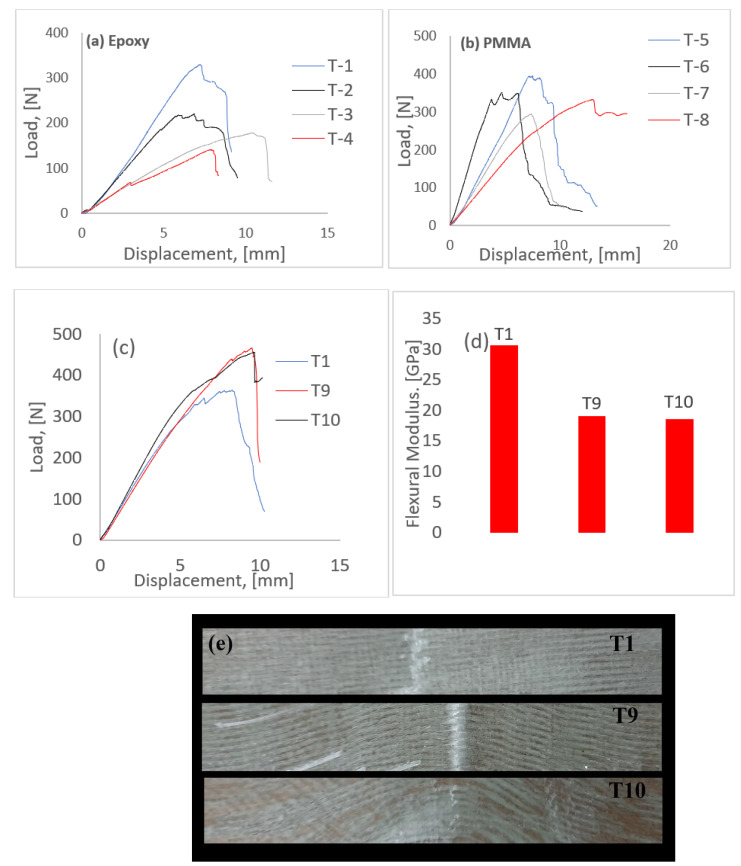
Load displacement curve from the flexural test. (**a**) Specimens with epoxy resin, (**b**) specimens with PMMA as resin, (**c**) specimens with in-plane waviness, (**d**) flexural modulus of specimens with in-plane waviness, and (**e**) specimens post-failure (in-plane waviness).

**Figure 4 polymers-14-01360-f004:**
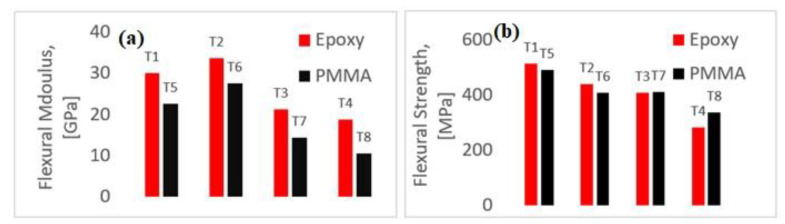
Flexural modulus and strength of pure glass, hybrids, and samples with waviness. (**a**) Flexural modulus comparisons from T-1 to T-8 and (**b**) Flexural strength comparison from T-1 to T-8.

**Figure 5 polymers-14-01360-f005:**
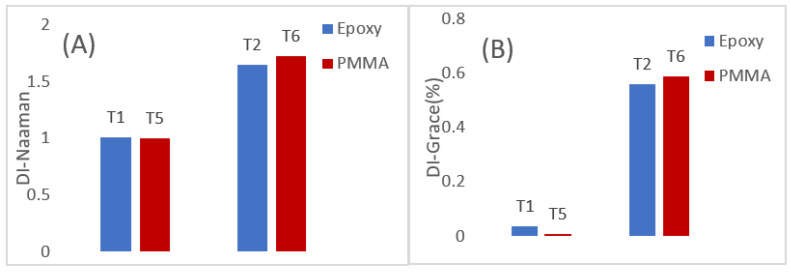
Ductility index (DI) based on the energy model developed by Naaman et al. [50] and Grace et al. [51], (**A**) DI as per Naaman, (**B**) DI as per Grace.

**Figure 6 polymers-14-01360-f006:**
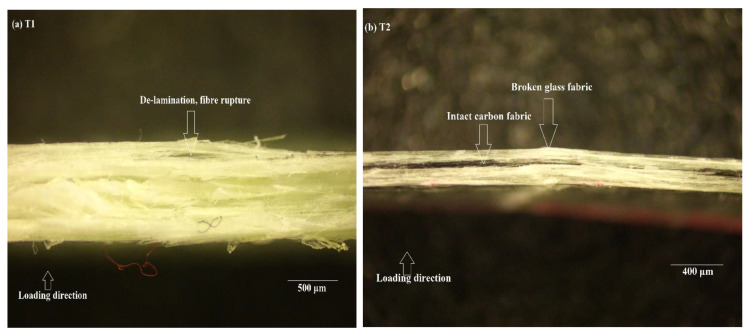
Magnified image of damaged cross-section of T1 (**a**) and T2 (**b**).

**Figure 7 polymers-14-01360-f007:**
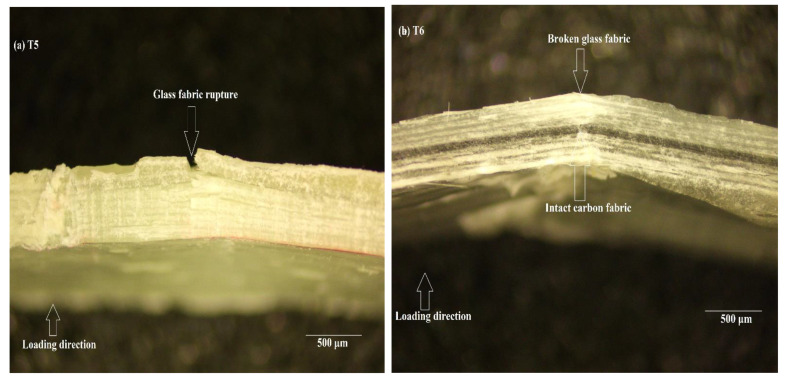
Magnified image of damaged cross-section of T5 (**a**) and T6 (**b**).

**Figure 8 polymers-14-01360-f008:**
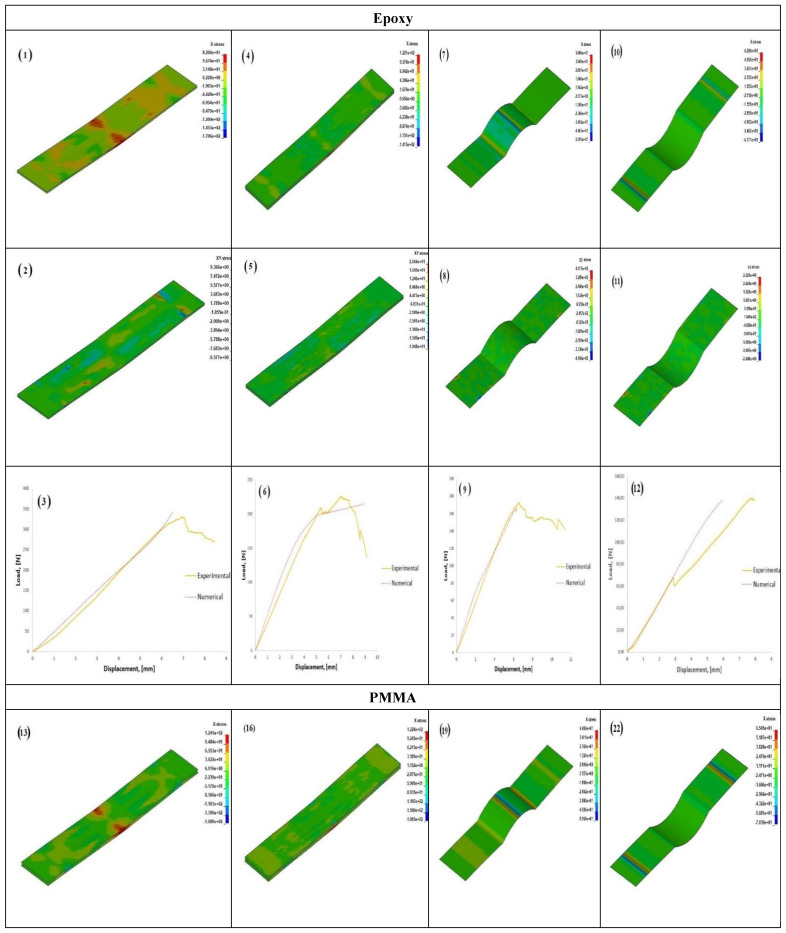
Numerical results (stress fields along the -x and -xydirections, load displacement plots) captured on LS DYNA for all the specimen types studied. (**1**) Stress in x-direction for T-1, (**2**) Stress in xy-direction for T-1, (**3**) Load curve for T-1. (**4**–**6**) Stresses and load curve for T-2. (**7**–**9**) Stresses and load curve for T-3. (**10**–**12**) Stresses and load curve for T-4. (**13**–**15**) Stresses and load curve for T-5. (**16**–**18**) Stresses and load curve for T-6. (**19**–**21**) Stresses and load curve for T-7. (**22**–**24**) Stresses and load curve for T-8.

**Figure 9 polymers-14-01360-f009:**
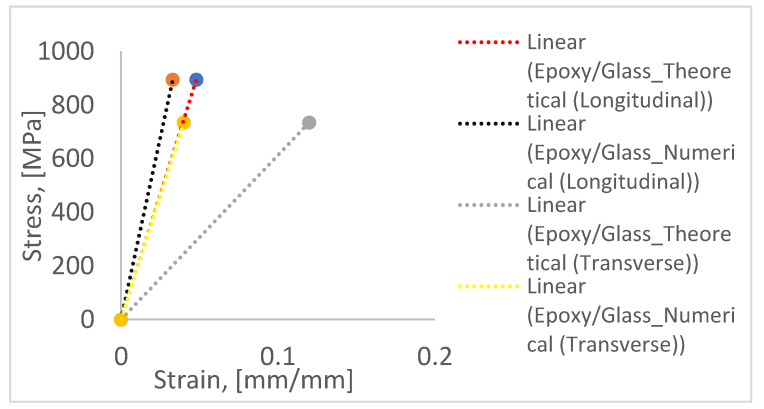
Theoretical and numerical strains for the parametric study.

**Figure 10 polymers-14-01360-f010:**
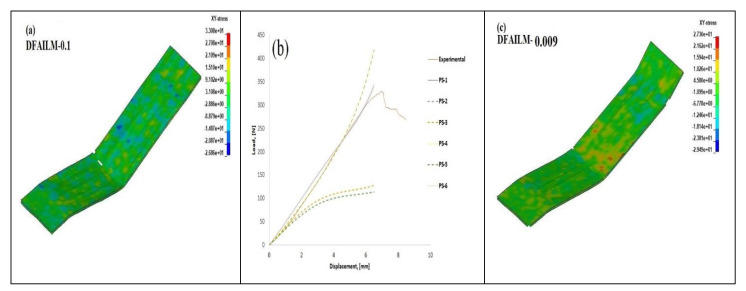
Parametric study on influence of DFAILT and DFAILM on flexural behaviour of T1. (**a**) XY-stress when DFAILM = 0.1. (**b**) Load-displacement plot for the six parametric studies listed in Table 4. (**c**) XY-stress when DFAILM = 0.009. (**d**) XY-stress when DFAILT = 0.1. (**e**) XY-stress when DFAILT = 0.033 and DFAILM = 0.04. (**f**) XY-stress when DFAILT = 0.009.

**Table 1 polymers-14-01360-t001:** Composite architecture.

Specimen Code	Symbol	Architecture	Fabric	Fibre Orientation	Resin
T-1	Glass  Carbon 	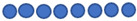	Glass	Uni-directional	Epoxy
T-2	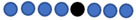	Glass and Carbon	Uni-directional	Epoxy
T-3/T-9	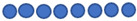	Glass	Uni-directional	Epoxy
T-4/T-10	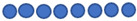	Glass	Uni-directional	Epoxy
T-5	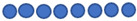	Glass	Uni-directional	PMMA
T-6	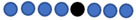	Glass and Carbon	Uni-directional	PMMA
T-7	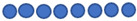	Glass	Uni-directional	PMMA
T-8	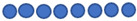	Glass	Uni-directional	PMMA

**Table 2 polymers-14-01360-t002:** Resin viscosity and density.

Resin Material	Property	Value	Units
Epoxy (Bisphenol A)	Viscosity at 25 °C	710	mPas
	Density at 25 °C	1.15	g/cm^3^
Epoxy (Bisphenol A)	Viscosity at 25 °C	14	mPas
	Density at 25 °C	0.94	g/cm^3^
PMMA	Viscosity at 20 °C	500	mPas
	Density at 20 °C	1	g/cm^3^

**Table 3 polymers-14-01360-t003:** Composite specimen properties for numerical modelling. *X_T_*: strength in tension (longitudinal); *X_C_*: strength in compression (longitudinal); *Y_T_*: strength in tension (transverse); *Y_C_*: strength in compression (transverse); *S_L_*: shear strength.

Resin Type	E_1_ (GPa)	E_2_ (GPa)	G_12_/G_13_ (GPa)	$\nu_{12}$	$X_{T}$ (MPa)	$X_{C}$ (MPa)	$Y_{T}$ (MPa)	$S_{L}$ (MPa)	$Y_{C}$ (MPa)
Epoxy/Glass	32.4	8.1	2.6	0.22	680	600	35	37	35
Epoxy/Carbon	63	40	9	0.16	709	473	501	146	199
PMMA/Glass	23.16	2.1	2.62	0.38	325	246	16	42	128
PMMA/Carbon	47	3	1.8	0.13	1300	882	15	40	120

**Table 4 polymers-14-01360-t004:** Comparison between experimental and numerical loads obtained in flexure.

Specimen Code	T-1	T-2	T-3	T-4	T-5	T-6	T-7	T-8
Experimental Load (N)	329.60	226.25	172.40	139.95	428.21	440.50	230.00	295.20
Numerical Load (N)	341.34	229.78	165.92	138.38	439.94	492.21	205.47	296.44

**Table 5 polymers-14-01360-t005:** Fibre and matrix mechanical properties.

Material	Young’s Modulus (GPa)	Tensile Strength (MPa)
Epoxy	3.2	70
Glass Fabric	81.0	2200

**Table 6 polymers-14-01360-t006:** Parameters adopted for the parametric study.

Parametric Study (PS)	DFAILT	DFAILM
**1**	**0.033**	**0.04**
2	0.048	0.04
3	0.009	0.04
4	0.1	0.04
5	0.033	0.009
**6**	**0.033**	**0.1**

## Data Availability

Not applicable.

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
