# Peer review of "Effect of Hybridization and Ply Waviness on the Flexural Strength of Polymer Composites: An Experimental and Numerical Study"

_polymers, 2022, doi:10.3390/polym14071360_

Round 1

Reviewer 1 Report

This paper by Sharath P. Subadra, et al show significant results , however there are some comments below :

  • PMMA resin should written in first appear in the paper then abbreviation can be written.
  • LS-DYNA software , details should highlight including company and origin .
  • Mesh type , should mentioned , square, triangle , etc .
  • Table caption in page 6 should written
  • PMMA give higher tensile, and displacement as shown in fig.2, deep discussion needs to write with supporting reference.
  • In one table compare between experimental and Numerical results.

Author Response

Dear Reviewer,

   Thank you for the review on my manuscript. I have addressed all the reviews on the draft, however on the 5th comment you gave I have the following issues.

1) The use of PMMA against epoxy gave me similar flexure behaviour even though a slight increase in load was seen but it was not very large.

2) The idea of PMMA in this article was to make it a more sustainable replacement for epoxy as it is much more easier to recycle. I have referenced these arguements in the article. In line with this, through experiments and numerical study we are trying to prove it behaves in a similar fashion as epoxy. 

As for the rest of the review I have addressed them and I am attaching a copy with this reply. 

Thank you

Sharath

Reviewer 2 Report

Dear Authors,

I attach comments in the file.

Yours sincerely,

Reviewer

Author Response

Dear Reviewer,

   Thank you for the review on our article. Based on the review you gave, I have addressed most of them. However on two two points, I would like to highlight my point of view.

24. Line 389-395. It seems to me that dependencies should be marked, as in the case of dependencies (1) and (2).

Here there are several dependencies and its quite difficult to compare and estimate them. Hence, for the article I have taken up just two parameters and the parametric studies where conducted based on these two parameters. And I have compared them in a table in the following section along with a plot. 

29. Table 3. There should be equal accuracy of the numbers.

This is quite difficult as the obtained values are based on numerical studies and its relation to the theoretical values arrived at on the fig  8.

And finally on the referencing, i have checked the correct template and corrected the first two referenced, the rest I will correct them in the final version which I will be submitting. 

I am attaching the corrected version with this reply. 

Thank you for the review,

Sharath

Round 2

Reviewer 1 Report

Authors reply to all comments 

Reviewer 2 Report

Dear Authors,

I hereby accept the corrections prepared by the authors.

Yours sincerely,

Reviewer